# Dirichlet-Swing: understanding spatio-temporal aspects of political elections in heterogeneous societies through agent-based simulation

**Adway Mitra** [ID]*

Indian Institute of Technology Kharagpur, Paschim Medinipur, West Bengal, India

* adway@ai.iitkgp.ac.in

## Abstract

Many countries have a system of electing members to their governing bodies through district-based elections. In each district, the party with maximum votes wins the corresponding "seat" in the governing body. However, the final seat distribution is strongly dependent on the geographical distribution of voters of different parties, and the party with most (or least) voters may not win the most (or least) number of seats if their voters are non-homogeneously distributed over the districts. This is further complicated in heterogeneous societies, where political preference of voters depends on their social identities, which is also related to their districts of residence. Projections of outcomes by sample surveys tend to fail in such situations. The aim of this paper is to explore how electoral outcomes are influenced by the geographical distribution of voters and community-centric voting preferences. We consider agent-based modeling of voters along with their locations, community memberships and voting preference. Our models represent the relations between these factors with their uncertainties through conditional probability distributions involving latent variables with Dirichlet Processes. Our models also represent spatio-temporal factors in elections – how geographical proximity between districts influence the voting preferences, and swing of votes across successive elections. We propose two novel models for vote swing between successive elections based on Dirichlet Processes, which is far more powerful than the existing models of Uniform Swing and Proportional Swing. For any choice of parameters, our models can be used to simulate a full election by Monte Carlo Sampling, and such simulations provide us a range of possible outcomes. We can also simulate surveys and study how their projections can deviate from the actual results. We discuss inference approaches to estimate the parameters to fit the model to actual district-based elections held in India.

**Data availability statement:** The data used here can be found in the official website of the Election Commission of India: https://www.eci.gov.in/statistical-reports. In case the above link does not work, here is the link to the data extracted from the above website for the 4 specific elections that have been considered in this paper (GJ17, GJ22, WB19, WB21). https://zenodo.org/records/18207734. The relevant file in this repository is IndiaElection_WB_GJ.xlsx The names of the parties have been anonymized.

**Funding:** This study was supported by Indian Institute of Technology Kharagpur in the form of a salary for A.M. The specific roles of this author are articulated in the 'author contributions' section. The funders had no role in study design, data collection and analysis, decision to publish, or preparation of the manuscript.

**Competing interests:** The author has declared that no competing interests exist.

# 1 Introduction

Many countries have a system of electing members to their governing bodies like parliament through district-based elections. In this system, the country is geographically divided into districts, and voters cast votes in their respective districts of residence. They vote in favor of any of the local candidates, who may belong to political parties. A common democratic setup is the district-based system in which the country is spatially divided into a number of regions called districts (or constituencies). There is a seat in the governing body corresponding to each district. The residents of each district elect a representative from a set of candidates, according to any voting rule (e.g., approval, ranked choice etc). In many countries, these candidates are representatives of political parties, and electors may cast their votes in favour of the parties rather than individual candidates. The winning candidate(s) is/are determined according to a scoring rule (e.g., plurality, Borda count etc). The winning candidate's party is considered to have won the corresponding seat in the governing body. The election results are understood in terms of the number of seats won by different parties, rather than the total number of votes obtained by them.

The relation between relative popularity of the different parties (as reflected by their aggregate vote shares) and the number of seats won by them in an election is a crucial and puzzling issue in Political Sciences, as analyzed by [1]. If the relative popularity of the different parties is spatially homogeneous across all the districts, then the most popular party may win all the seats. But this is very rarely the case. One reason for this may be the individual popularity of candidates may vary across districts, which may influence the voting decisions more than popularity of the parties. But a more complex reason is the spatial variation of demography across the country, since the popularity of different parties often varies with demography [2]. Demographics vary spatially as people usually prefer to choose residences based on social identities, such as race, religion, language, caste, profession and economic status. This process is sometimes called "ghettoization," where people with similar social identities huddle together in geographical regions [3,4]. Such spatial heterogeneity plays a very important role in district-based elections if different political parties represent the interests of different social groups. Even if a political party is not popular overall, it can win a few seats if its supporters are densely concentrated in a small number of districts, which forms strongholds of the party. On the other hand, a party which is overall quite popular, may fail to win many seats if its supporters are spread all over without concentration. Also, electors often vote according to the advice of local community leaders and other local factors [5], which causes "polarization" of voters in favour of one/two parties inside each district. There are relatively few statistical models for simulation of district-based elections. Eggenberger and Polya used the concept of Polya's urn to propose a statistical voting model, which simulates the effect that if one candidate gets a vote, there are likely to get more [6]. There have been attempts to extend these to multiple districts [7]. Another popular approach is Mallow's Model, which assumes a 'central' ranking over the candidates, and simulates individual votes by perturbing it. The impact of spatial distribution of voters on district-based elections have been studied by [8–10] in an analytical framework.

Though the aim of these works is to study *gerrymandering* (i.e., changing district boundaries to favor a party), they add a model to simulate the geographical distribution of voters. The work [1] studies *misrepresentation ratio*, a measure of distortion in the outcome due to the geographical distribution of voters. In this study, some simple models of probabilistic election simulation are used to back their theoretical results.

When successive elections are held in a demographic polity over regular intervals, they may or may not produce similar results. Outcomes of elections may change as the overall popularity of the parties may change due to contemporary factors. It can also happen that a party loses popularity within some segments of the population, but gains popularity in other segments. The change of popularity between successive elections is known as *vote swing*. One well-known model of vote swing is *uniform swing*, which assumes that the swing is similar in all the districts, which may result in a surge in favor of a party all over the country. Another model is *proportional swing*. The work [11] lays down an axiomatic definition of swing models, and consider alternative models based on a function that relates the swing in each district to the overall swing. Another work [12] focuses at voter level, as it tries to predict the behavior of *swing voters* based on factors influencing the voting decisions at the last minute.

Surveys are often carried out to forecast the election results. These surveys may be conducted by various agencies before or after the election. Usually a survey involves a small sample of the electorate, based on whose responses the vote share of the different parties is estimated. The number of seats to be won by the different parties can be estimated as well from this sample. However, the accuracy of these estimates depends on how well these samples represent the entire population. For example, the chosen samples may cover only a few districts, or misrepresent the true vote share of the different parties. This may arise either due to practical constraints (such as the difficulty of reaching certain geographical areas) or due to malicious intent or partisan bias of the survey agency. Furthermore, if a party is popular among some communities but unpopular among other communities, and the communities are unevenly distributed across districts, then such surveys will find it very difficult to predict the results accurately. These issues give rise to an important question: given a particular election, how likely is a particular survey to project the correct outcome in terms of seat distribution?

A significant amount of research work exists in predicting the election results from a survey under different conditions. Most of these works like [13–17] focus on finding the minimum number of samples needed by a survey to forecast the winner and/or the margin of victory with a given confidence, and efficient algorithms for the same. [18] extends this analysis to district-based settings, and provides algorithms to carry out the survey over a limited number of districts and a limited number of persons in each district. However, none of these works, to the best of our knowledge, predict the number of seats won by the parties in either deterministic or probabilistic way. One of the few works which attempts an alternative statistical approach based on regression and stratified sampling to forecast election results based on surveys is [19], who also apply their framework to forecast the Indian General Election 2019 using surveys conducted through Amazon Mechanical Turk in addition to socio-economic data of voters and their voting preferences. However, this work uses the Uniform Swing theory to convert their vote share projections to seat share projections.

One of the few research works using agent-based modeling for elections [20], where ABMs are used to forecast election outcomes. In this work, the authors use various attributes like age, gender etc of voters to predict their votes. The work also considers live experiments for election forecasting, where surveys are augmented by agent-based models. However, this work is not aimed at district-based elections.

The aims of this work are four-fold. First of all, we attempt to estimate the possible number of seats won by different parties. Secondly, we study the spatial correlation of results across districts, and attempt to incorporate it in our models. Third, we study the spatial variation of swing in results across successive elections, and provide a novel treatment for it. Our final aim is to evaluate the above for actual district-based political elections held in India, for which we need to solve a parameter estimation problem, and we propose ways of doing so.

Our approach depends heavily on the simulation of election outcomes. Recently, there have been attempts to systematically represent various aspects of district-based elections through voter-centric agent-based statistical models [21–23]. In this work, we utilize some of these models to simulate complete election results, by considering every elector's vote as a latent random variable. We aim to capture different voting trends seen in societies through these models. When it comes to vote swings in elections, uniform swing and proportional swing are unable to explain the situation when a party loses votes overall, but manages to win a few new seats. We present models based on Dirichlet Processes for this purpose. Our next target is to study election surveys and projection of results based on them. We build upon the model to simulate surveys developed by [24], and carry out further analysis on the limitations of survey strategies to predict election results. Our simulations suggest an important result – it is generally more sample-efficient to estimate the swing with respect to the past election than to directly estimate the results of the current election.

## 2 Notations and problem definition

We consider district-based 1-plurality elections, i.e., the candidate/party with maximum votes in a district wins the corresponding seat. Consider $N$ voters divided among $S$ districts as $\{N_1, \ldots, N_S\}$. There are $K$ parties in fray, each of whom has a candidate in each district. Denote by $Z$ the **complete election**, where $Z = \{Z_1, Z_2, \ldots, Z_S\}$ where $Z_s = \{Z_{s1}, \ldots, Z_{sK}\}$ denotes the total number of votes of the parties in district $s$.

For each district $s$, we denote by $\theta_s = \{\theta_{s1}, \ldots, \theta_{sK}\}$ as the local vote share. Clearly, $\theta_{sk} = \frac{Z_{sk}}{N_s}$. Again, $\theta = \{\theta_1, \ldots, \theta_K\}$ denotes the overall vote shares of the parties, where $\theta_k = \frac{\sum_k Z_{sk}}{N}$.

Denote by $U_s$, the winning party in district $s$, and by $W_k$, the number of districts where the candidate from party $k$ is the winner. Clearly, $\sum_k W_k = S$. Finally, denote by $X$: the actual electoral outcome. It has two parts: $X = \{X^1, X^2\}$ where $X^1 = \{\theta_1, \ldots, \theta_K\}$, and $X^2 = \{\frac{W_1}{S}, \ldots, \frac{W_K}{S}\}$, i.e., the vote shares and seat shares of the parties.

An election is defined by $Z$, since the overall vote share and seat share of all parties can be easily calculated from it. $Z$ is a combinatorial structure, as each $Z_s$ specifies a $K$-way partition of $N_s$. However, not all partitions are equally likely (for example, it is very unlikely that all voters in a district vote for the same party). An election simulation model considers $Z$ as a random variable, and attempts to specify a probability distribution over it. But since it is difficult to define a distribution over such a complex structure, it can be implicitly at voter level, through random variable $V_{si}$ which indicates the vote by the $i$-th voter of district $s$. This is the approach of *Agent-based Modeling*, where each voter is considered as an agent. Clearly, $Z_{sk} = \sum_{i=1}^{N_s} I(V_{si} = k)$, where $I$ denotes the indicator variable. One of our aims in this paper is to understand the distribution over $Z$ that is represented by the models, and to compare it with actual district-based elections in India. This analysis can establish how realistic our models are. Another aim is to study the distribution of the election outcome $X$ (which is a function of $Z$) as induced by the simulation model.

Denote by $Y$: the projected results based on the surveys, which also has two parts: $\{Y_1, Y_2\}$ which are the projected vote shares and seat shares of all the parties. A survey model simulates $\hat{Z}$ from $Z$, by drawing samples of voters from a random subset of the districts and querying their votes. $Y$ is obtained by estimating the vote and seat shares from $\hat{Z}$ and extrapolating them to the whole electorate. However, $Y$ can be very different from $X$, if $\hat{Z}$ is not a good representative of $Z$. Another aim of this paper is to study the distribution of $Y$ under sampling strategies conditioned on $Z$, i.e., to study how likely a survey is to predict the correct outcome of a particular election.

A schematic diagram of the entire process flow is provided in Fig 1.

## 3 Agent-based models

We will now discuss a series of agent-based models to simulate voter behavior. These models proceed by assigning political preferences and locations to each voter by sampling from suitable probability distributions, and using these to simulate the election.

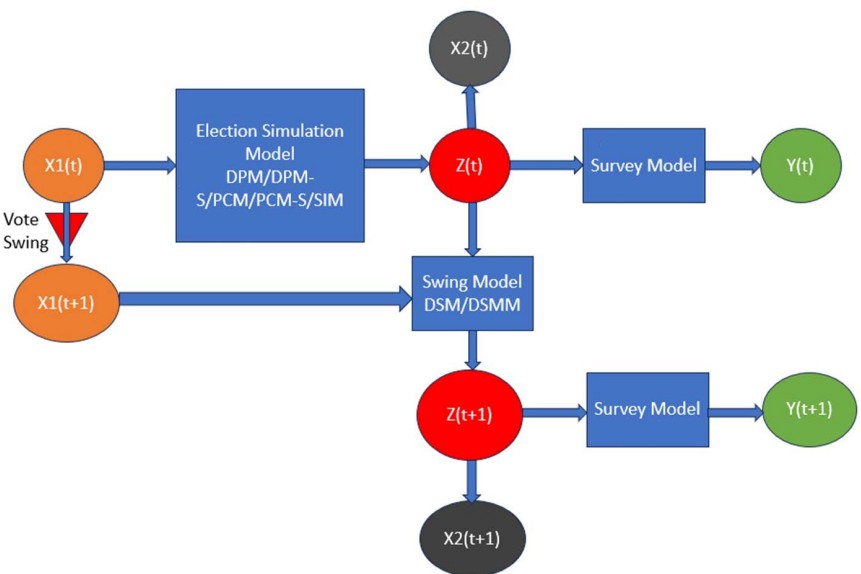

**Fig 1. A schematic diagram of the process flow in this paper, showing the different types of models, including their inputs and outputs.**

## 3.1 Spatial distribution models

Earlier works like [21,22,25] introduced some agent-based voter models like the District-wise Polarization Model (DPM) and Partywise Concentration Model (PCM). The DPM model aims to represent the fact that many voters prefer to vote based on local factors, rather than overall popularity of the different parties. In DPM model, each vote is simulated by sampling from either the overall or the local popularity (vote share) of the parties, which is decided by a Bernoulli distribution with a concentration parameter $\alpha$. High value of $\alpha$ suggests more importance of local popularity than overall, which allows parties which are less popular to win some seats. Low value of $\alpha$, on the other hand, tends to preserve the overall popularity trends in each district, hence the most popular party is likely to win most seats.

$$prob(V_{si} = k) \propto (\alpha_s \theta_{sk} + (1 - \alpha_s)\theta_k) \tag{1}$$

The PCM model, also discussed in [22,25] aims to represent the geographical variation in concentration of voters of each party. Here, each voter of each party $k$ is stochastically assigned to a district, based on the logic that they are more likely to be assigned to a district where that party is already popular, but this likeliness is again controlled by a concentration parameter $\eta_k$. This is supposed to represent the phenomena that supporters of a party tend to be residing closely, as political partisanship often depends on demographics or community membership. So in this model, we do not simulate $V_{si}$ directly, but rather sample the vote $V_i$ for voter $i$ based on overall vote shares, and then assign them to a district $D_i$ as discussed above.

$$prob(V_i = k) \propto \theta_k$$
$$prob(D_i = s|V_i = k) \propto (\eta_k \theta_{sk} + (1 - \eta_k)U(1, K)) \tag{2}$$

The power of this model is that $\eta_k$ is specific to parties, so different parties can have different levels of concentration, thereby allowing a wider range of possibilities on $Z$. In general, high concentration is useful for parties with low vote share as it enables them to win seats, but parties with higher vote share may win more seats if their votes are less concentrated spatially.

It may be noted that the conditional distributions of both the above models are related to the famous Chinese Restaurant Process [26], which is a derivative of Dirichlet Processes. It follows the logic of *rich-getting-richer*, i.e., a voter is more likely to vote for a candidate who is already popular (in case of DPM), or likely to reside in a district where there are already many voters of the same party (in case of PCM).

### 3.2 District geography model

When two districts are geographically aligned, studies like [27] show that their voting patterns are quite similar. Accordingly, we aim develop models that consider that the districts are geographically located, and for each district there is a set of neighboring districts. We re-cast the DPM and PCM models in this scenario. In case of DPM model, we consider that the popularity of different parties in each district is influenced not just by their overall popularity, but also by the popularity in the neighboring districts. For any district $s$, $\phi(s)$ denotes the districts that are geographical neighbors of $s$. We add an extra parameter $\beta$, which indicates the influence of the neighboring districts. Accordingly, we add another term to the conditional distribution of the DPM model:

$$prob(V_{si} = k) \propto \alpha_s \theta_{sk} + \beta_s(1-\alpha_s)\theta_{\phi(s)k} + (1-\alpha_s)(1-\beta_s)\theta_k \qquad (3)$$

Here, $n_{\phi(s)k}$ indicates the average share of votes for party $k$ in the neighboring districts of $s$. Clearly, both $\alpha, \beta \in (0, 1)$. Also, high value of $\beta$ indicates a stronger influence of neighboring regions, indicating that a party's votes in any district will be more correlated with its votes in the neighboring districts. We call this new model as Geography-augmented District Polarization Model (GDPM).

In case of PCM model, when the assignment of voters to districts is dictated not only by the popularity of parties in those districts, but also to neighboring districts. Once again, we add a new parameter $\beta$ to include the impact of neighboring districts into the probability of assigning voter $i$ to district $s$. We add a term to the conditional distribution that is related to the average vote share of the same party $k$ in the neighboring districts $\phi(s)$ of $s$.

$$prob(V_i = k) \propto \theta_k$$
$$prob(D_i = s|V_i = k) \propto \eta_k \theta_{sk} + \beta_k \theta_{\phi(s)k} + (1-\beta_k)(1-\eta_k)U(1, K) \qquad (4)$$

Once again, a high value of $\beta$ indicates that voters of any party tend to reside in nearby districts. This version of the model will be called as Geography-augmented Partywise Concentratation Model (GPCM).

### 3.3 Social identity model

Apart from geographical proximity, another major factor that is known to influence voting patterns is community membership. Studies like [27] have shown that people belonging to the same social community tend to exhibit similar voting patterns, even if they belong to districts that are geographically far apart. Now, we present another model where the social communities and their political preferences are directly parameterized. Assume that there are $C$ social communities, and $\kappa_c$ denotes the proportion of the electorate from community $c$. To every voter $i$, we assign their community as $C(i) \sim Categorical(\kappa)$ ($\kappa = \{\kappa_1, \ldots, \kappa_C\}$). Voters from the same community tend to reside in the same district. Voter $i$ is assigned to district $D(i)$ by following a Chinese Restaurant Process [26] with parameter $\eta_c$. Voter $i$ from community $C(i) = c$ is assigned to district $s$ with probability proportional to $\eta_c \sum_{j=1}^{i-1} I(C(j) = c)I(S(j) = s)$ (i.e., number of voters from same community as $i$ already residing in district $s$), or to any district chosen uniformly at random with probability proportional to $(1 - \eta_c)$. This ensures that for each community, certain districts turn into strongholds. Note that this is very similar to PCM, with the exception that we are considering community memberships instead of political preference in assigning voters to districts.

 

Each community is associated with a prior over the political preferences of its members. For community $c$ and party $k$, we assign $\Phi_{ck} \in \{-1, 0, 1\}$, indicating if the relation between them is bad ($-1$), neutral (0) or good (1). Also, a variance $\sigma_k$ is associated with each party (which may be drawn from a Gamma prior with parameters specific to the party). Finally, for each voter $i$, their valuation of party $k$ is denoted by $\lambda_{ik} \sim \mathcal{N}(\Phi_{ck}, \sigma_k)$ where $c = C(i)$. A party with high $\sigma$ is strongly liked by some voters and strongly disliked by other voters across communities (indicating its "polarizing" nature), but for a party with low value of $\sigma$, most members of each community have similar values. Clearly, this valuation $\lambda_{ik}$ can be either positive or negative. While these valuations definitely influence each voter's voting choice, voters may also get influenced their social network. The $i$-th voter combines their own valuations $\lambda_{ik}$ with the mean valuations of other voters in the same district, as $\hat{\lambda}_{ik} = \mu\lambda_{ik} + (1-\mu)\bar{\lambda}_{ik}$ where $\bar{\lambda}_{ik} = \frac{\sum_{j=1}^{N} I(S(j)=S(i))\lambda_{jk}}{\sum_{j=1}^{N} I(S(j)=S(i))}$, and $\mu \sim Beta(a, b)$. This local influence is independent of community affiliation. Finally, voter $i$ casts their vote $V_i$ in favor of the party for which their moderated valuation $\bar{\lambda}_i$ is maximum.

In a nutshell, the election model may be written as:

$$C(i) \sim Categorical(\kappa)\forall i \in \{1, N\}$$
$$D(i) \sim CRP(C, \eta)\forall i, \sigma_k \sim Gamma(\gamma_k)\forall k$$
$$\lambda_{ik} \sim \mathcal{N}(\phi_{ck}, \sigma_k) \text{ where } c = C(i), \forall i, k$$
$$\mu \sim Beta(a, b), \hat{\lambda}_{ik} = \mu\lambda_{ik} + (1-\mu)\bar{\lambda}_{ik}\forall i, k$$
$$\text{where } \bar{\lambda}_{ik} = \frac{\sum_{j=1}^{N} I(D(j) = D(i))\lambda_{jk}}{\sum_{j=1}^{N} I(D(j) = D(i))}$$
$$V_i = argmax_k\bar{\lambda}_{ik}$$

$$(5)$$

It may be noted that while DPM and PCM require the overall vote share $\theta$ ($X^1$) and concentration parameter $\alpha$ or $\eta$ as input, the SIM takes parameters $\kappa, \eta, \phi, \sigma$ as the inputs to generate the full result $Z$. From this, both $X^1$ and $X^2$ can be easily calculated.

Once again, we can consider a Geography-augmented version of the SIM (GSIM), where the assignment of voter $i$ to any district $s$ considers not only the number of members of the same community in district $s$, but also in the neighboring districts. This encourages the members of each community to reside in geographical clusters, which may span multiple districts. Once again, this is realistic in most societies.

A summary of all of these models is provided in Fig 2.

## 4 Modeling vote swing across elections

Elections are not one-time events, but are held regularly, at regular intervals. When we analyze one election, it is natural to compare and contrast the results of one election with the next – which party gained/lost how many votes and/or seats. The change in vote shares $\theta(t)$ and $\theta(t + 1)$ between successive elections is referred to as *swing*. We denote it by $\Delta\theta(t) = \theta(t + 1) - \theta(t)$.

Clearly, the above definition is related to change in overall vote share across the country. However, to understood how this translates to changes in seat distribution, we need to understand the swings at district level, as denoted by $\Delta\theta_s(t) = \theta_s(t + 1) - \theta_s(t)$. The question is, how is $\Delta\theta_s(t)$ related to $\Delta\theta(t)$? Clearly, $\Delta\theta(t) = \sum_s \frac{n_s}{N}\Delta\theta_s(t)$, and hence $E(\Delta\theta_s(t)) = E(\Delta\theta(t))$. As already mentioned, the most common model assumes uniform swing, i.e., $\Delta\theta_s(t) = \Delta\theta(t)$. However, as will be illustrated in the Experiments section, in many elections we find that a party improves its vote share in some districts even as it loses votes overall. This cannot be explained by either the Uniform or the Proportional Swing Theories. In this work, we have proposed two models of swing.

| Model | Input | Output | Aim | Parameter |
|---|---|---|---|---|
| DPM | Vote share X1 | Full result Z, seat share X2 | Each voter chooses between local and global popularities of parties | Concentration α |
| DPM-S | Vote share X1 | Full result Z, seat share X2 | Each voter chooses between local, regional and global popularities of parties | Concentration α, geographical influence β |
| PCM | Vote share X1 | Full result Z, seat share X2 | Voters of each party are spread over the districts, voters of same party may prefer same districts | Partywise concentration η1, ....., ηK |
| PCM-S | Vote share X1 | Full result Z, seat share X2 | Voters of each party are spread over the districts, voters of same party may prefer same or neighboring districts | Partywise concentration η1, ....., ηK, geographical influence β |
| SIM | Community share Θ, community-party relation φ | Vote share X1, Full result Z, seat share X2 | Each voter is influenced by their community's relationship with different parties, as well as local and global popularities of the parties | Community parameter κ, district concentration η, social influence μ |
| DSM | X1(t), Δ, Z(t) | Z(t+1), X2(t+1) | Distribute the overall swing in vote share of parties (compared to past election) to swings across the districts | Dirichlet parameters (ν1, ν2, ...... νK) for vote swing |
| DSMM | X1(t), Δ, Z(t) | Z(t+1), X2(t+1) | Distribute the previous votes of each party in previous election across all parties | Dirichlet parameters (ρ11, ρ12, ...... ρKK) for vote swing from each party to others |

**Fig 2. A summary of the different models proposed in this paper.**

### 4.1 Dirichlet swing model

First, we consider a Dirichlet Process Mixture Model [28] for vote swing. We note that $\sum_k \Delta\theta_k(t) = 0$, as the gain in vote share of some parties is offset by loss of the others, and $\sum_k \theta(t) = 1$. So there is no standard probability distribution that can be used to model $\Delta\theta(t)$. However, if we make the assumption that $|\Delta_k\theta(t)| < \frac{1}{K}$, i.e., no party's vote share swings by more than $\frac{1}{K}$, then we can consider a new variable $\hat{\Delta}_k\theta(t) = \Delta_k\theta(t) + \frac{1}{K}$. Here, $\hat{\Delta}\theta(t)$ lies on the $(K-1)$-simplex, i.e., it can be considered as a $K$-categorical Probability Mass Function. Naturally, we can model $\hat{\Delta}\theta(t)$ with a Dirichlet Distribution.

Specifically, we consider a swing prior $\hat{\Delta}\theta_0(t)$ which follows a Dirichlet Process with a base distribution $H$ and parameters $\gamma_0$. We choose $H$ to be a Dirichlet Distribution with parameters $\nu = \{\nu_1, \ldots, \nu_K\}$. Hence, every *atom* drawn from this base distribution is a $K$-categorical PMF, and $\hat{\Delta}\theta_0(t)$ is a discrete mixture distribution over such *atoms*. The weights of each atom is obtained by a stick-breaking process (GEM) with parameter $\gamma_0$. The swing in each district is one of these atoms, drawn from $\hat{\Delta}\theta_0(t)$.

$$\hat{\Delta}\theta_0(t) \sim DP(H, \gamma_0)$$
$$\hat{\Delta}\theta_s(t) \sim Categorical(\hat{\Delta}\theta_0(t))$$
$$\theta_s(t+1) = \theta_s(t) + \Delta\theta_s(t), \text{ where } \Delta\theta_{sk}(t) = \hat{\Delta}\theta_{sk}(t) - \frac{1}{K}$$

(6)

Note that the overall swing $\Delta\theta(t)$ is not explicitly included in the model, but it can be calculated from the district-wise swings.

The hyperparameters $\nu$ and $\gamma_0$ are very vital in this construct. A low value of $\gamma_0$ indicates that the stick-breaking process will attach small weights to each atom. So when samples are drawn from the mixture distribution $\hat{\Delta}\theta_0(t)$, they are likely to

be all distinct, i.e., different districts will have different swings. On the other hand, a large value of $\gamma_0$ suggests that one atom can have a large weight, i.e., most districts will have the same swing, which is similar to the Uniform Swing Theory. The Dirichlet hyperparameters $v$, on the other hand, are estimators of the overall swing, since $E(\hat{\theta}_k(t)) = E(\hat{\theta}_{sk}(t)) = \frac{\nu_k}{\sum_j \nu_j}$. However, it can be shown that high magnitudes of $\nu_k$ promote low variance of $\hat{\theta}_{sk}(t)$, i.e., most district-wise swings will be similar. But low values of $\nu_k$ promote high variance of $\hat{\theta}_{sk}(t)$, suggesting the different districts can have very different swings.

## 4.2 Dirichlet swing matrix model

Next, we consider a more sophisticated model based on the swing matrix, that accounts for how the voters of party $k$ in the previous election, voted in the current election. In other words, we attempt to parameterize the quantity $M_{skl}(t) = p(V_{si}(t + 1) = l | V_{si}(t) = k)$. We define a transition matrix $M_s(t)$ of size $K \times K$, where each row is a PMF corresponding to the voters of party $k$ in the previous election. For any party $k$, we first define a latent prior $M_k^0$, based on which the party's transition matrix in any district $s$ is defined. Once again, this is modeled through a Dirichlet Process, as below:

$$M_k^0(t) \sim DP(H_k, \gamma_k) \forall k \in \{1, K\}$$
$$M_{sk}(t) \sim Categorical(M_k^0(t)) \forall s \in \{1, S\}, \forall k \in \{1, K\}$$
$$V_{si}(t + 1) \sim Categorical(M_{sk}(t)) \forall i \in \{1, n_s\} \text{ where } k = V_{si}(t) \tag{7}$$

Clearly, the base distribution $H_k$ is a Dirichlet distribution specific to party $k$, with parameters $\{\rho_{k1}, \ldots, \rho_{kK}\}$. Usually, $\rho_{kk}$ will be higher than the rest, as most voters of party $k$ in previous election are likely to stick to the same party in the current election too. Each atom drawn from the base distribution is a transition distribution for the previous voters of party $k$, and for each district $s$, we draw a one such atom from the mixture distribution $M_k^0(t)$. Once again, low magnitudes of $\rho$ promote high variance across the districts, while high magnitudes encourage similar behavior across all districts. This model is inspired by previous models like [29,30] that considers transition matrix of voter preferences, and develops maximum-likelihood estimates of this matrix. However, we not only extend this idea to district-based elections, but also propose a generative model for the transition matrix.

A summary of all of these models is provided in Fig 2.

# 5 Exploring the outcome space

Having defined the models, we now conduct a series of simulation studies based on these models. These simulations are done in an idealized, synthetic setting with $N = 10000000$, $S = 100$. The districts are considered to be of equal population, and arranged in the form of an uniform square grids with neighborhoods defined accordingly. We study the impacts of varying the different parameters on the outcomes in case of the different models.

## 5.1 Vote share vs seat share

In this experiment, we explore how a certain vote share can translate into seat shares. For this, we consider 3 different values of vote shares for $K = 2$, $K = 3$ and $K = 4$, and run the DPM and PCM under different settings of their parameters $\alpha$ and $\eta$ to obtain the seat shares. The numbers reported are averaged over 10 runs over each setting, and the standard deviation is also reported. The results are shown in Table 1. We find that while the most popular party (highest vote share) can win nearly all the seats in some settings, it may be able to win just above half the seats in some other settings, if the voters of the parties are spatially more concentrated, as indicated by higher values of $\alpha$ and $\eta$.

For the GDPM and GPCM models, for suitable choice of the spatial coherence parameter $\beta$, the seat distribution over parties may remain the same as those of DPM and PCM, but their geographical distribution changes, as we see adjacent

Table 1. Exploring space of outcomes (Seat shares) for different vote shares using DPM and PCM models with different concentration parameter settings.

| Model | Param | $X^1_1$ | $X^1_2$ | $X^1_3$ | $X^1_4$ | $X^2_1$ | $X^2_2$ | $X^2_3$ | $X^2_4$ |
|---|---|---|---|---|---|---|---|---|---|
| DPM-1 | $\alpha = 0.8$ | 0.6 | 0.4 | x | x | 1.0 | 0 | x | x |
| DPM-2 | $\alpha = 0.9$ | 0.6 | 0.4 | x | x | 0.84 | 0.16 | x | x |
| DPM-3 | $\alpha = 0.95$ | 0.6 | 0.4 | x | x | 0.73 | 0.27 | x | x |
| DPM-4 | $\alpha = 0.8$ | 0.45 | 0.35 | 0.2 | x | 0.94 | 0.06 | 0 | x |
| DPM-5 | $\alpha = 0.9$ | 0.45 | 0.35 | 0.2 | x | 0.74 | 0.23 | 0.03 | x |
| DPM-6 | $\alpha = 0.95$ | 0.45 | 0.35 | 0.2 | x | 0.58 | 0.3 | 0.12 | x |
| DPM-7 | $\alpha = 0.8$ | 0.4 | 0.3 | 0.2 | 0.1 | 0.97 | 0.03 | 0 | 0 |
| DPM-8 | $\alpha = 0.9$ | 0.4 | 0.3 | 0.2 | 0.1 | 0.75 | 0.2 | 0.04 | 0.01 |
| DPM-9 | $\alpha = 0.95$ | 0.4 | 0.3 | 0.2 | 0.1 | 0.55 | 0.26 | 0.14 | 0.05 |
| PCM-1 | $\eta = [0.5, 0.95]$ | 0.6 | 0.4 | x | x | 0.82 | 0.18 | x | x |
| PCM-2 | $\eta = [0.95, 0.5]$ | 0.6 | 0.4 | x | x | 0.72 | 0.28 | x | x |
| PCM-3 | $\eta = [0.95, 0.95]$ | 0.6 | 0.4 | x | x | 0.73 | 0.27 | x | x |
| PCM-4 | $\eta = [0.5, 0.95, 0.95]$ | 0.45 | 0.35 | 0.2 | x | 0.71 | 0.24 | 0.05 | x |
| PCM-5 | $\eta = [0.95, 0.5, 0.95]$ | 0.45 | 0.35 | 0.2 | x | 0.54 | 0.39 | 0.07 | x |
| PCM-6 | $\eta = [0.95, 0.95, 0.95]$ | 0.45 | 0.35 | 0.2 | x | 0.59 | 0.34 | 0.07 | x |
| PCM-7 | $\eta = [0.5, 0.5, 0.95, 0.95]$ | 0.4 | 0.3 | 0.2 | 0.1 | 0.92 | 0 | 0.08 | 0 |
| PCM-8 | $\eta = [0.5, 0.95, 0.95, 0.95]$ | 0.4 | 0.3 | 0.2 | 0.1 | 0.71 | 0.22 | 0.07 | 0 |
| PCM-9 | $\eta = [0.95, 0.95, 0.95, 0.95]$ | 0.4 | 0.3 | 0.2 | 0.1 | 0.57 | 0.3 | 0.11 | 0.02 |

seats having similar vote shares and same winners more frequently than in the original models. In Fig 3, we show the winner maps for two such elections over $S = 100$, one using DPM and the other with GDPM, for the same vote share $X^1 = [0.45, 0.35, 0.2]$. We can observe higher spatial coherence in the second election.

In case of the SIM model, both the vote share and seat share are generated. So we run the experiments for different number $C$ of communities, their proportions $\eta$ and different community-party relations $\Phi$. We consider four scenarios – two involving $C = 3$ communities, and two more involving $C = 5$ communities. In case of $C = 3$, we set the community proportions as $\eta = \{0.5, 0.3, 0.2\}$, i.e., one large, medium and small community. For $C = 5$, their proportions are set to $\eta = \{0.35, 0.35, 0.1, 0.1, 0.1\}$, i.e., two large and three small communities. We constrain $\varphi$ such that for each party $k$, $\sum_{c=1}^{C} \eta_c \phi_{ck} \leq 0.5$, i.e., we assume that a party cannot satisfy many persons without dissatisfying some others. In each case, Scenario 1 (polarized) involves Party 1 that is favored by the larger communities and opposed by the smaller ones, Party 2 that is favored by the smaller communities and opposed by the larger ones, and Party 3 which is neutral to all communities. The third party, however has $\sigma = 2$, higher than the other two with $\sigma = 1$ indicating that it has strong individual supporters and opponents. These relations are represented by $\phi^1$. In Scenario 2 (non-polarized), each party is favored by one or more communities, but not opposed by the rest. One party again has high $\sigma = 2$, the others have $\sigma = 1$. These relations are represented by $\phi^2$. We report the resulting vote shares and seat shares in Table 2, along with standard deviation of votes across seats. Once again, we report the numbers over 10 runs of experiments in each setting. It is seen that in polarized scenario of $\phi_1$, the neutral party fails to win any seat with fewer communities, but can do well with more communities involved. Also, with more communities involved, there is very less difference between $\phi_1$ and $\phi_2$. Local influence is found to benefit the parties that support the larger communities and harms the centrist party, particularly when fewer communities are involved.

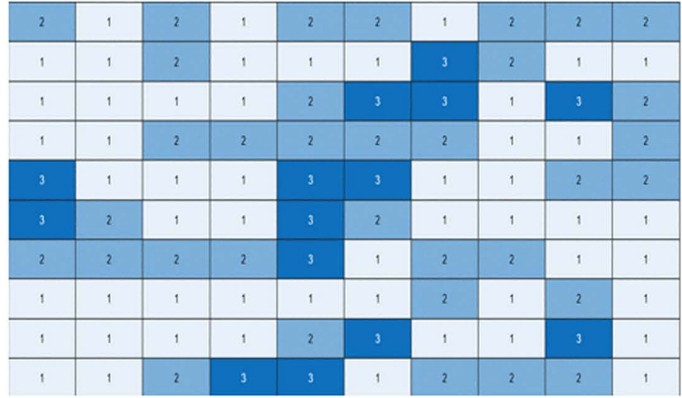
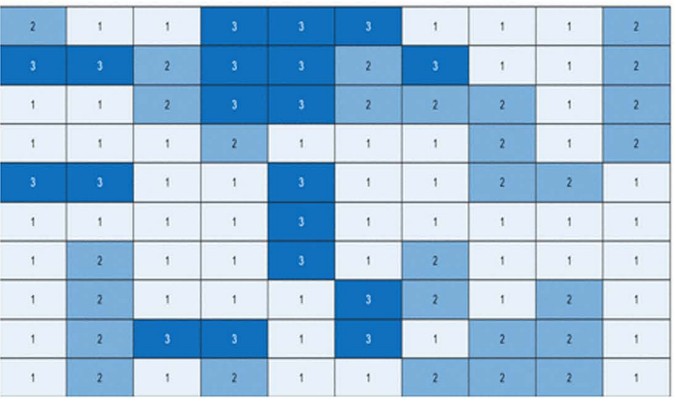

Winner map for DPM                Winner map for GDPM

**Fig 3. Winner maps of two elections over 100 seats, simulated by DPM (left) and GDPM (right).** The vote share is same in both cases $[0.45, 0.35, 0.2]$ and also the seat share $[0.52, 0.34, 0.14]$.

**Table 2. Social identity model in two scenarios $\phi^1$, $\phi^2$, for $C = 3$ and $C = 5$. Above: individual preference, Below: local influence variants of SIM.**

| Model | Param | $X_1^1$ | $X_2^1$ | $X_3^1$ | $X_1^2$ | $X_2^2$ | $X_2^2$ |
|-------|-------|---------|---------|---------|---------|---------|---------|
| SIM-1 | $\phi^1(C = 3)$ | 0.35 | 0.34 | 0.31 | 0.49 | 0.35 | 0.16 |
| SIM-2 | $\phi^2(C = 3)$ | 0.37 | 0.38 | 0.25 | 0.42 | 0.40 | 0.18 |
| SIM-3 | $\phi^1(C = 5)$ | 0.33 | 0.36 | 0.31 | 0.35 | 0.30 | 0.35 |
| SIM-4 | $\phi^2(C = 5)$ | 0.33 | 0.36 | 0.31 | 0.35 | 0.31 | 0.34 |
| SIM-5 | $\phi^1(C = 3)$ | 0.36 | 0.34 | 0.3 | 0.47 | 0.31 | 0.22 |
| SIM-6 | $\phi^2(C = 3)$ | 0.43 | 0.35 | 0.22 | 0.45 | 0.33 | 0.22 |
| SIM-7 | $\phi^1(C = 5)$ | 0.35 | 0.32 | 0.33 | 0.38 | 0.24 | 0.38 |
| SIM-8 | $\phi^2(C = 5)$ | 0.35 | 0.32 | 0.33 | 0.37 | 0.24 | 0.38 |

### 5.2 Vote swing vs seat swing

Next, we study the process of swings between successive elections. For each of the elections simulated by DPM, PCM, SIM in the previous analysis (referred to as DPM-1, PCM-2 etc in Tables 1 and 2), we simulated the *next election* using both the Dirichlet Swing Model (DSM) and the Dirichlet Swing Matrix Model (DSMM). We consider different values of the swing $\Delta\theta(t)$. The base distribution parameters $v$ is chosen as $B\Delta\hat{\theta}(t)$, where hyperparameter $B$ controls the variance across the atoms, i.e., district-wise swings. In each case, we estimate the new vote share and the corresponding seat share. The detailed results are shown in Table 1 in the Supporting Information.

An example is illustrated in Fig 4, in an election over $S = 100$ seats, where the first election is simulated by DPM, and second elections by DSM using $B = 1$ and $B = 10$ based on the first election. For $B = 1$, we see a different winner in 54 seats, while for $B = 10$ only 33 seats change hands. Similar results are also found for DSMM. We illustrate the results graphically in Fig 5. We plot the change is vote share of party $k$, i.e., $\theta_k(t) = X^1(k, t + 1) - X^1(k, t)$ against its corresponding change in seat share, i.e., $X^2(k, t + 1) - X^2(k, t)$. We plot this separately for each party $k = \{1, 2, 3\}$ (denoted by 'o', '+' and '*' in the figure). $X^1(t + 1), X^2(t + 1)$ are estimated using DSM with different parameter settings (denoted by different colors).

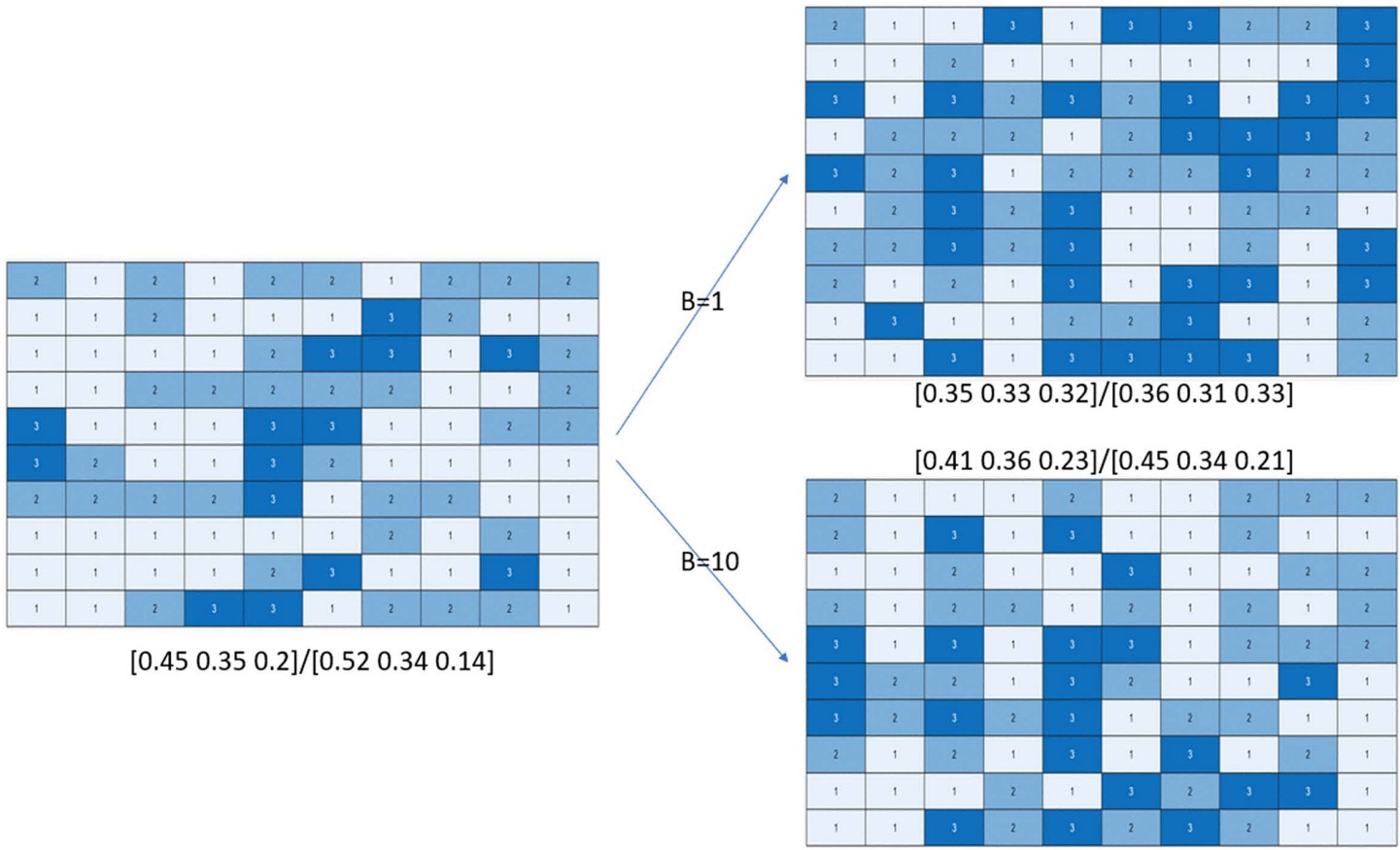

**Fig 4. Swing from previous election (left) to next election (right) under Dirichlet Swing Model, with parameter** $B = 1$ **(top right) and** $B = 10$ **(bottom right).**

We find that for low values of $B$ the seat swing can be very significant and even counter-intuitive, for example in some cases we find Party 1 gaining 3% votes but still losing seats on average. Such swings are more moderated for higher values of $B$. Furthermore, we vary the Dirichlet Base Distribution parameters for both DSM and DSMM, which regulates the variance of swings across the districts.

## 6 Simulating Indian elections with parameter estimation

Next, we consider political elections held in India, which is known as the largest parliamentary democracy in the world. The aim is to examine if these theoretical models can simulate these elections under appropriate parameter settings. We study individual elections as well as swings across consecutive elections. Elections are district-based, based on plurality voting, which suits the framework considered here. Also, elections are held at regular intervals of time, either for the state assemblies or for the national parliament, and vote swing between successive elections is a politically significant phenomena which is studied by many political scientists. For this study, we consider 4 state-level elections held in India in recent times. The details of the election are given in Table 3.

### 6.1 Simulation by models

The first experiment is to compare model simulations with the actual election results. There are two aims –

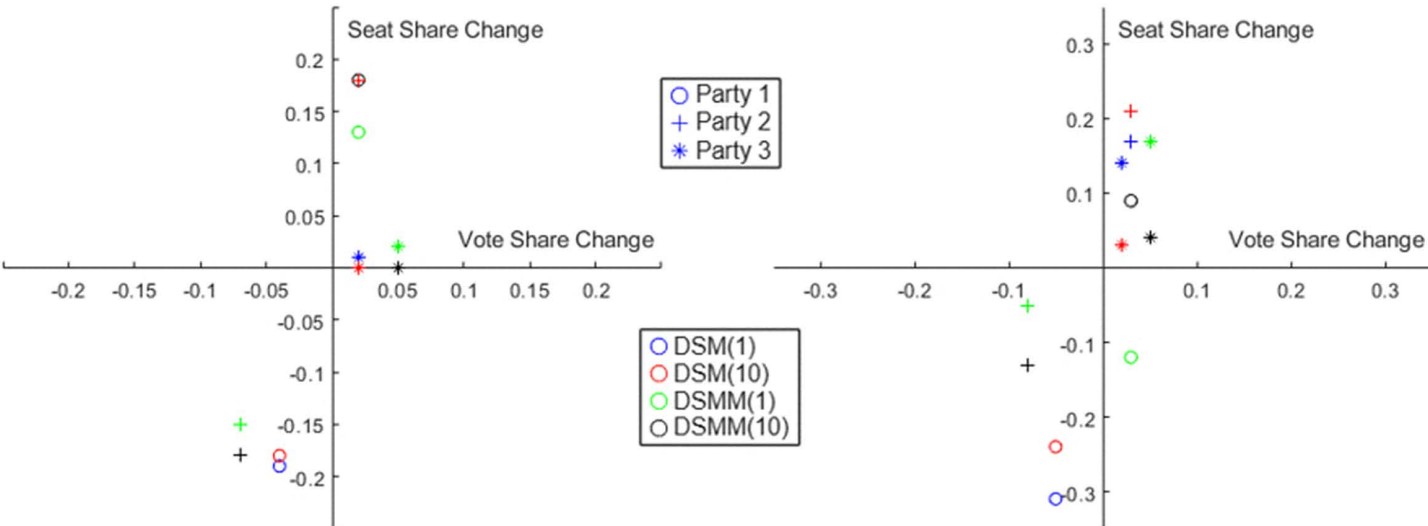

**Fig 5. Vote Share Swing vs Seat Share Swing for different parties under different different swing models and parameters.** Left panel: Districtwise Swing Matrix Model (DSMM), Right Panel: Districtwise Swing Model (DSM). Circle 'o' denotes Party 1, Plus '+' denotes Party 2, Star '*' denotes Party 3. Each color (blue, red, green, black) indicates a different parameter setting.

**Table 3. Details of a few past elections from different states in India.**

| Election | State | Year | N | S | $X^1_1$ | $X^1_2$ | $X^1_3$ |
|---|---|---|---|---|---|---|---|
| GJ17 | Gujarat | 2017 | 30251142 | 182 | 0.49 | 0.42 | 0.09 |
| GJ22 | Gujarat | 2022 | 29580492 | 182 | 0.56 | 0.3 | 0.14 |
| WB19 | West Bengal | 2019 | 55158913 | 294 | 0.45 | 0.13 | 0.42 |
| WB21 | West Bengal | 2019 | 57373983 | 294 | 0.51 | 0.09 | 0.4 |

1. *Calibration for Validation*: verify if the proposed models can simulate the election results (seat share) given the vote shares under suitably chosen parameters

2. *Counterfactual Analysis*: to explore what could have been the results of these elections had the voters of the different parties been geographically distributed in a different way. In other words, we aim to see the impact of the district boundaries on the results.

We use the DPM, GDPM, PCM and GPCM models for these analyses. We cannot use the SIM model, since we do not have community-wise information about either residence or voting preferences.

In the first experiment, we consider the seat share of different parties in each of the elections, for both optimal and two standard parameter settings. The optimal parameter setting is chosen by grid search, however there are other approaches to finding the optimal values of the parameters (discussed later). The two standard parameter settings include one with low values of the concentration parameters ($a$ for DPM and $\eta$ for PCM) – which suggest spatially homogeneous distribution of each party's voters, and another with high values of these parameters, suggesting strong spatial heterogeneity, and encouraging each party's supporters to be spatially concentrated. Table 4 shows the results for DPM and PCM. Apart from the seat share, we also indicate the standard deviation of votes received by each party across the districts, in both the actual and the simulated elections. We find that under the optimal parameter settings, both DPM and PCM models

**Table 4. Simulation of seat shares in 4 Indian elections by DPM (middle) and PCM (below) under optimal and default parameter settings.** The actual seat shares in these elections are shown in the upper part of the table. Simulated results in case of optimal parameter settings match the actual results in all cases.

| Election | Opt. Param. | | | Def. Param. 1 | | | Def. Param. 2 | | |
|---|---|---|---|---|---|---|---|---|---|
| | $X_1^2$ | $X_2^2$ | $X_3^2$ | $X_1^2$ | $X_2^2$ | $X_3^2$ | $X_1^2$ | $X_2^2$ | $X_3^2$ |
| GJ17 | 0.53 | 0.44 | 0.03 | x | x | x | x | x | x |
| GJ22 | 0.88 | 0.09 | 0.03 | x | x | x | x | x | x |
| WB19 | 0.54 | 0.05 | 0.41 | x | x | x | x | x | x |
| WB21 | 0.75 | 0.01 | 0.24 | x | x | x | x | x | x |
| GJ17 | 0.53 | 0.42 | 0.05 | 0.71 | 0.29 | 0.0 | 0.49 | 0.43 | 0.08 |
| GJ22 | 0.88 | 0.1 | 0.02 | 0.99 | 0.01 | 0.0 | 0.61 | 0.26 | 0.13 |
| WB19 | 0.56 | 0.02 | 0.42 | 0.59 | 0.01 | 0.4 | 0.46 | 0.12 | 0.42 |
| WB21 | 0.71 | 0.03 | 0.26 | 0.84 | 0 | 0.16 | 0.52 | 0.09 | 0.39 |
| GJ17 | 0.53 | 0.44 | 0.03 | 0.97 | 0.03 | 0.0 | 0.53 | 0.44 | 0.03 |
| GJ22 | 0.86 | 0.10 | 0.04 | 1.0 | 0.0 | 0.0 | 0.68 | 0.26 | 0.06 |
| WB19 | 0.53 | 0.03 | 0.44 | 0.86 | 0.0 | 0.14 | 0.5 | 0.06 | 0.44 |
| WB21 | 0.73 | 0.01 | 0.26 | 1.0 | 0.0 | 0.0 | 0.58 | 0.03 | 0.39 |

can recreate the actual results. We also see from the 2 default parameter settings that the results could have been very different, had the voters been distributed differently.

The optimal choice of the parameters, to fit a given model to actual election results may be determined in various ways. The simplest approach, used in this work is grid search over the parameter space. However, this approach can be quite inefficient, especially for models like PCM with multiple parameters, and for elections with a large number of voters. Possible other alternative approaches include Bayesian Optimization [31] and Simulation-based Inference [32,33], where the main idea is to explore the parameter space efficiently, as we need to run the simulation with each candidate parameter value and compare the result with the observations. The general idea is to first run trial simulations using a few parameter values from a prior distribution, identify those which produce results similar to the observations, and search further around them till a good match with the observations are obtained. There are also neural network-based surrogates, which can make simulation-based inference more efficient by predicting the outcomes instead of running the full simulations at each candidate value [34–37]. Finally, the recently developed paradigm of Differentiable Agent-based Modeling [38–40] such as GradABM allows us to implement the aforementioned Agent-based Models as differentiable functions, whose parameters can be estimated through back-propagating the error between the simulated results and observations.

## 6.2 Geographical coherence

The next experiment examines the question of geographical coherence – i.e., do adjacent districts have similar vote shares and winners? This question has been examined by some earlier studies like [30]. We evaluate this in both the actual and the simulated elections. In any state of India, the districts are numbered in such a way that any two consecutively numbered districts are geographically adjacent to each other. First of all, we compare the vote shares $\theta_s$ and $\theta_{s+1}$ between every pair of consecutively numbered (neighboring) districts using Kullback-Leibler (K-L) Divergence. Similarly, we compare the vote share $\theta_s$ and $\theta_{s'}$ where $s'$ is chosen randomly from among the districts that are not neighbouring to $s$. We define a measure Vote Spatial Coherence (VSC) as the ratio of the average K-L Divergence between neighboring and non-neighbouring districts are thus compared, as $\frac{mean(KL(s,s'))}{mean(KL(s,s+1))}$. A high value of this ratio suggests that the vote shares in two non-neighboring districts are less similar than those in two neighboring districts. A corollary of the strong correlation of vote shares between adjacent seats is that, adjacent seats are often won by the same party. The next analysis is to check the winners. We calculate a Winner Coherence Score (WSC) as $\sum_s I(V_s = V_{s+1})$ for both real and simulated elections.

The results are illustrated in Table 5. We find that both VSC and WSC are quite high for the elections that we have considered. We repeat the same analysis for the elections simulated by DPM, GDPM, PCM and GPCM, and it is found that GDPM and GPCM definitely increases the spatial concentration in terms of both vote share and winner compared to DPM and PCM. While WSC scores from the simulated elections tend to match the actual election, the VSC tends to be overestimated.

## 6.3 Simulation of swings

The third experiment in this section is to analyze the swing between successive elections. We consider two pairs of successive elections in India – (Gujarat2017 vs Gujarat2022) and (Bengal2019 vs Bengal2021). In each case, we start with the full results $Z(t)$ of the earlier election, as well as the vote shares $\theta(t+1)$ in the later election. Using these, we try to predict the results $Z(t+1)$ of the later election. In this case, prediction means not only estimation of seat shares, but also predicting the winner in each district (since we already know the winner of each district in the earlier election). Using both the DSM and DSMM, we estimate the results in each district. From $\theta(t+1)$ we calculate $\hat{\Delta}\theta(t)$, which is used as the parameters of the base distribution $H$, multiplied by a constant $B$. High values of $B$ encourage low variance, i.e., similar swings in each district, and its low values encourage high variance, i.e., different swings in different districts. We calculate the following:

- In how many seats does each party register an increase in vote share? (denoted by Gain-1, Gain-2 etc). This is measured as $Gain_k(t) = \sum_s I(\theta_{sk}(t+1) > \theta_{sk}(t))$. This can be calculated for both actual and simulated values of $\theta(t+1)$.

- How many seats change hands from one party to another? This is measured by $Flip(t) = \sum_s I(U_s(t+1) \neq U_s(t))$, and can be calculated in both actual and simulated elections.

- In how many seats are the winners correctly predicted (Accuracy)? Note that, for our Swing models, the results for each district varies from one simulation to another, as these are random variables. Hence it is futile to calculate the seat-wise accuracy in each run of the simulation. Instead, we run the simulations 10 times, and the winning party for each seat is noted in each simulation. The probabilities of each party winning a particular seat is calculated accordingly, and the accuracy is calculated using these probabilities. Hence we measure $Acc(t) = \sum_s Prob(\hat{U}_s(t) = U_s(t))$, where $U$ is the actual winner in district $s$ and $\hat{U}$ is the winner according to simulations by a model.

The results are provided in Table 6. For both Gujarat (GJ17) and West Bengal (WB19), we run simulations to predict GJ22 and WB21 respectively, using both DSM and DSMM using different values of $B$. We can actually make maximum-likelihood estimates of $B$ (using Minka's algorithm for estimating Dirichlet parameters [41]) from the actual results of GJ22 and WB21, which turn out to be $B = 10$ and $B = 18$ respectively. Clearly, in case of DSM, smaller value of $B$ indicates larger swings of vote shares and flipping of winners across the districts. We also calculate the fraction of districts where each party improves its vote share, i.e., $X_k^1(s, t+1) > X_k^1(s, t)$, denoted by Gn1, Gn2, Gn3. We find that DSMM model outperforms DSM in predicting the results of the second election in both cases, and in terms of both vote shares

**Table 5. Comparing spatial correlations between vote shares (VSC) and winners (WSC) in actual elections and elections simulated by DPM, GDPM, PCM, GPCM.** The simulated results that are closest to the actual results are highlighted.

| Election | Actual | | DPM | | GDPM | | PCM | | GPCM | |
|---|---|---|---|---|---|---|---|---|---|---|
| | VSC | WSC | VSC | WSC | VSC | WSC | VSC | WSC | VSC | WSC |
| GJ17 | 1.2 | 0.6 | 0.99 | 0.45 | **1.2** | 0.5 | 1.07 | 0.55 | 1.33 | **0.6** |
| GJ22 | 1.48 | 0.78 | 0.86 | 0.78 | 5.3 | 0.8 | 0.87 | 0.78 | **1.25** | 0.77 |
| WB19 | 3.8 | 0.73 | 1.05 | 0.47 | 12.4 | 0.57 | 1.00 | 0.47 | 1.92 | **0.69** |
| WB21 | 1.4 | 0.8 | 1.0 | 0.62 | 14.8 | 0.68 | 0.98 | 0.6 | 2.27 | **0.8** |

**Table 6. Comparing swings in vote and seat shares across successive elections in reality and as simulated by DSM and DSMM.** Upper 5 rows are for swings across GJ17 and GJ22, while the bottom rows are for swings across WB19 and WB21. The results of GJ17 are $X^1(t) = [0.49, 0.42, 0.09]$, $X^2(t) = [0.53, 0.44, 0.03]$, and those of WB19 are $X^1(t) = [0.44, 0.42, 0.14]$, $X^2(t) = [0.54, 0.41, 0.05]$.

| Swing(B) | $X^1(t+1)$ | $X^2(t+1)$ | Flips | Gn1 | Gn2 | Gn3 | Acc |
|---|---|---|---|---|---|---|---|
| Reality | [0.56,0.30,0.14] | [0.88,0.09,0.03] | 0.45 | 0.87 | 0.13 | 0.62 | – |
| DSM(1) | [0.51,0.27,0.22] | [0.52,0.22,0.26] | 0.58 | 0.49 | 0.23 | 0.26 | 0.49 |
| DSM(10) | **[0.55,0.29,0.15]** | [0.77,0.16,0.07] | 0.39 | 0.67 | **0.14** | 0.6 | 0.72 |
| DSMM(1) | [0.57,0.31,0.12] | [0.77,0.19,0.04] | 0.29 | 0.91 | 0.05 | **0.62** | 0.72 |
| DSMM(10) | [0.58,0.3,0.12] | **[0.93,0.05,0.02]** | **0.4** | **0.95** | 0.01 | 0.84 | **0.84** |
| Reality | [0.51,0.4,0.09] | [0.75,0.25,0] | 0.3 | 0.85 | 0.34 | 0.13 | – |
| DSM(1) | [0.46,0.36,0.18] | [0.46,0.34,0.2] | 0.55 | 0.47 | **0.36** | 0.35 | 0.46 |
| DSM(10) | [0.49,0.39,0.12] | [0.6,0.35,0.05] | 0.34 | 0.62 | 0.39 | 0.35 | 0.65 |
| DSM(18) | [0.5,0.39.0.11] | [0.63,0.33,0.04] | **0.3** | 0.67 | 0.39 | 0.31 | 0.68 |
| DSMM(1) | **[0.5,0.4,0.1]** | [0.64,0.33,0.03] | 0.16 | 0.93 | 0.37 | 0.07 | 0.76 |
| DSMM(10) | **[0.5,0.4,0.1]** | **[0.7,0.29,0.01]** | 0.19 | **0.93** | 0.28 | **0.1** | **0.81** |

and seat shares. This is particularly true when we use $B = 10$ as the parameter of DSMM model. However, DSMM tends to underestimate the vote share swings and winner-flips across the districts.

A graphical illustration of the above results is provided in Fig 6, where we plot the errors between the actual results and the results simulated by both the DSM and DSMM models for both the Indian states mentioned above. The errors are calculated with respect to both the vote share and seat share. Specifically, if the actual vote share and seat shares are denoted by $X^1_a(t+1), X^2_a(t+1)$, and those simulated by a model are denoted by $X^1_m(t+1), X^2_m(t+1)$, then we compare $e^1_m = X^1_m(t+1) - X^1_a(t+1)$ and $e^2_m = X^2_m(t+1) - X^2_a(t+1)$. Once again, we show the errors separately for each party (denoted by 'o','+','*') and each model with different parameter settings (denoted with different colors). We see that in both cases, the seat share errors are quite high for DSM with $b = 10$ (blue color), but for DSMM these errors are much less.

## 7 Simulation of election surveys

The aim of a survey is to estimate the underlying reality by examining a small number of samples. In this case, the underlying reality is $Z$, and the aim of the survey is to predict the vote shares $X^1$ and seat shares $X^2$. This is obtained by selecting a small subset of the voters and finding out their preferences (it is assumed that they respond truthfully) from which a projection $Y = \{Y^1, Y^2\}$ is made. Since this is a theoretical work, we cannot carry out actual surveys, and hence we aim to simulate surveys on real or simulated elections. For this purpose, we consider the survey model presented in [24]. While this paper explored the $p(X|Y)$, i.e., possible actual outcomes given a survey projection, here we analyze $p(Y|X)$, i.e., possible projections by survey of an election.

### 7.1 Uniform and stratified sampling

First we consider the survey model of [24], which simulates uniform sampling. The main question here is, how to choose these respondents. As already discussed, voting preferences may vary from district to district. While it may not be possible to cover all districts, an unbiased survey can be considered to choose a few districts uniformly at random, and also choose respondents uniformly at random from these districts. This approach of *Uniform Sampling* has been discussed by other works like [18], which provided lower bounds on the fraction of districts to be sampled, and the number of people to be queried in each district to be able to predict the winner correctly. In our model, we represent these as parameters $f_s$ and $f_n$. We further assume that the number of people are queried in each chosen district is proportional to the number of voters in that district.

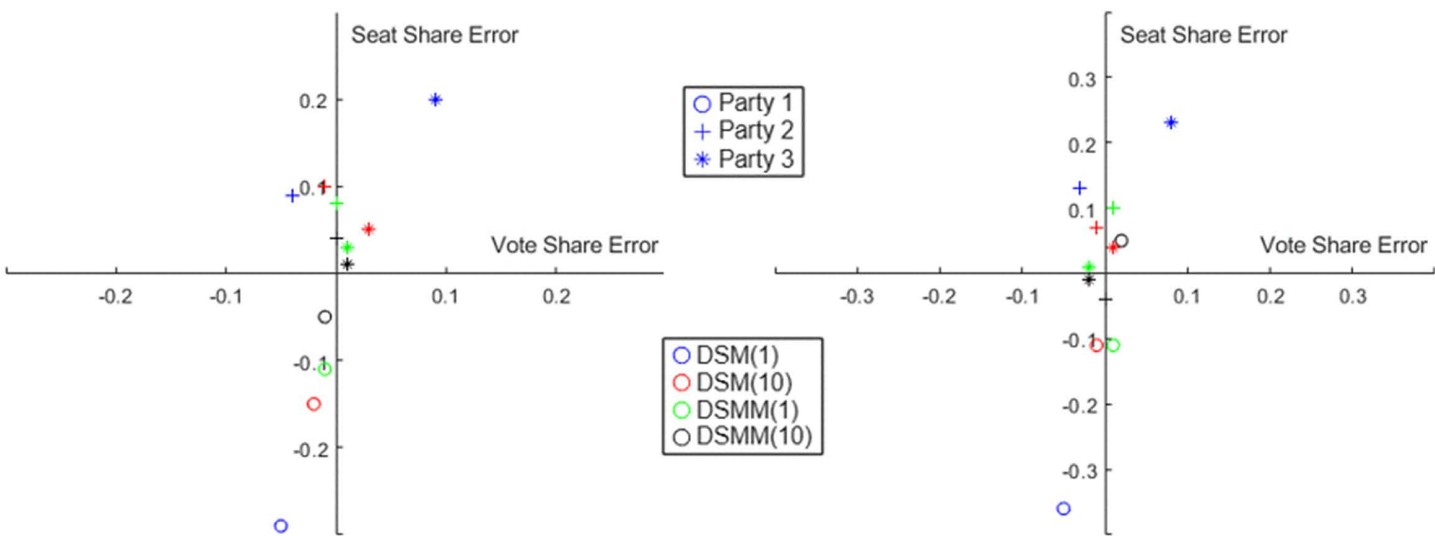

**Fig 6. Comparing the error in predicted Vote Share (X-axis) and prediced Seat Share Swing (Y-axis) for different parties under different swing models and parameters.** Left panel: Predictions of West Bengal Elections 2021 (WB21) based on West Bengal Elections 2019 (WB19). Right panel: Predictions of Gujarat Elections 2022 (GJ22) based on Gujarat Elections 2017 (GJ17). Circle 'o' denotes Party 1, Plus '+' denotes Party 2, Star '*' denotes Party 3. Blue and Red colors indicate predictions by DSM model, while Green and Black colors indicate predictions by DSMM model.

The survey model mentioned above does not consider community identities of the respondents. An alternative survey model is *Stratified Sampling*, where the respondents are first chosen according to a proportion $\hat{\eta}_s$ of communities in district $s$. $\hat{\eta}$ may be different from $\eta$ either due to the surveyors' lack of knowledge about the social structure, or systemic biases. Thus, the numbers of respondents from different communities are $\{c_{s1}, \ldots, c_{sC}\} \sim Mult(\{N_s f_n, (\hat{\eta}_{s1}, \ldots, \hat{\eta}_{sC})\})$, and these respondents can be queried. This can be simulated using the $(V, C, S)$ variables of the SIM.

### 7.2 Projection of results

Suppose in district $s$, a survey finds $\{n_{s1}, \ldots, n_{sK}\}$ respondents in favour of the $K$ parties. Clearly, this follows a Multinomial Distribution with parameters $\{N_s f_n, (\theta_{s1}, \ldots, \theta_{sK})/N_s\}$. The next question is, given the survey results, how to project the outcome $\{Y^1, Y^2\}$. Our model estimates the total vote share by simply aggregating the number of respondents across all districts, who expressed preferences for different parties. In other words, $Y^1(k) = \frac{\sum_s n_{sk}}{Nf_n}$ ($Nf_n$ is the total number of respondents) for party $k$. Next, in each of the $Sf_s$ districts where we carried out the survey, we identify the party with maximum number of votes among the respondents from that district. Thus, we find the number of districts $\{v_1, \ldots, v_K\}$ "won" by the different parties, and we use this as our estimate $Y^2$ of the overall seat share, i.e., $Y^2(k) = \frac{v_k}{Sf_s}$. We call this the **Direct Projection** approach.

We next consider another alternative approach: to estimate the swing with respect to the previous election. For this, we can consider the Dirichlet Swing Matrix Model, where the aim is to estimate the transition matrix. Here, each respondent is queried on the party they voted for in the current and the previous election, from which we can estimate the swing matrix in each of the queried district, where $\hat{M}_{skl}(t) = \frac{\sum_i I(V_{si}(t)=k)I(V_{si}(t+1)=l)}{\sum_i I(V_{si}(t)=k)}$. Though these estimates may be used to estimate the vote share in those particular districts, they do not say much about the remaining districts. We use these estimated transitions to make Maximum Likelihood estimates of the Base Dirichlet hyperparameters. Using the properties of Dirichlet Distribution, it can be derived easily that the Maximum Likelihood estimate of $\rho_{kl}$ is $M_{skl}$. Using these parameters, we carry

out simulations using the Dirichlet Swing Matrix Model, which gives us a projected outcome. We can carry out a number of such simulations and project the mean value of their outcomes. We call this approach as **Swing Projection**.

We discuss some measures for comparing projected results $Y^2$ and actual results $X^2$. We do not compare $X^1$ and $Y^1$, as uniform sampling is likely to estimate the vote shares correctly.

First of all, we consider the **Manhattan Distance** between $X^1$ and $Y^1$, which is given by $d_M(X^2, Y^2) = \sum_k |X_k^2 - Y_k^2|$. We calculate the mean value of this quantity over $G$ runs of a particular survey strategy and settings, $d_M(X^2, Y^2) = \frac{1}{G} \sum_{i=1}^{G} \sum_k |X_k^2 - Y_{ik}^2|$, where $\{Y_1^2, \ldots, Y_G^2\}$ are the projected outcomes of these surveys.

Another measure we consider is, how likely is a survey to project accurate results? We consider $Y^2$ to be an accurate estimate of $X^2$ if $\sum_k |X_k^2 - Y_k^2| < \delta$. We are interested in the quantity $prob(\sum_k |X_k^2 - Y_k^2| < \delta)$, which is approximated as $\frac{1}{G} \sum_i I(\sum_k |X_k^2 - Y_{ik}^2| < \delta)$.

### 7.3 Direct survey vs swing survey

The aim of this section is to compare the different survey strategies discussed above. We simulate surveys on elections simulated by the aforementioned models like DPM and SIM, and also on actual elections from the Indian election dataset. The main aims of the experiments is to estimate how likely the different strategies are to project accurate results, under different values of $(f_n, f_s)$ parameters. In our experiments, we consider $G = 100$.

In the first experiment, we consider the importance of spatial coverage $f_s$ and person coverage $f_n$. In general, we can expect that if there is significant diversity in terms of vote share across the districts, performance should improve if we consider more districts in our survey. But if such diversity does not exist, then surveying more districts (high $f_s$) has no advantage. Similarly, if the vote shares of different parties are close to each other then sampling more voters (high $f_n$) can improve the estimates, but this does may not be true if the vote shares are well-separated. We consider i) four elections simulated by the GDPM model with different levels of popular support and the concentration parameter $\alpha$, ii) four elections simulated by G-PCM with different levels of popular support and the concentration parameter $\eta$, iii) 2 elections simulated by SIM (one where different communities have comparable preferences, and one where different communities have markedly different preferences). The results are illustrated in Figs 7–9 respectively, while the full results in tabular form are provided in the Supporting Information. We find that there is no straightforward relation between projection performance and $f_n$ or $f_s$. Performance tends to improve with $f_s$ in case of the elections simulated by DPM and PCM, but less so in case of SIM. The reverse is true in case of $f_n$. The projection performance generally tends to be significantly worse in case of the elections simulated by SIM, as it is a more sophisticated model capable of adding more layers of uncertainty through community-based preferences of voters. The same experiment is repeated for the 4 Indian elections, and the results are shown in Table 7. In all cases we find that increasing district coverage $f_s$ is more effective than increasing person coverage $f_n$. We also notice that for GJ-17 and WB-19, where the first two parties had quite close vote/seat shares, the probability of successful projection was below 40%, but this was much higher in GJ-17 and WB-21 when the first party had a big lead over the rest.

The aim of the second experiment is to compare direct projections with swing-based projections. The hypothesis is that, if we can estimate the vote swings with respect to the previous election from surveys, we can achieve better projection accuracy with a certain number of respondents, than trying to directly estimate the current election's result. We perform this experiment on two Indian elections too – Gujarat Assembly Elections of 2017 and 2022, and West Bengal Elections in 2019 and 2021. We simulate surveys in the new elections on different values of $(f_s, f_n)$, to make projections for them. These are compared with the direct projections based on the simulated surveys on the second election. Once again, we calculate both Manhattan distance and Projection Accuracy, and compare them in Table 8. We use $B = 10$ for West Bengal and $B = 6$ for Gujarat. We find that in most cases, using Swing-based projection gives better results in terms of both Manhattan Distance and Probability of Accurate Projection, especially if the district coverage is lower.

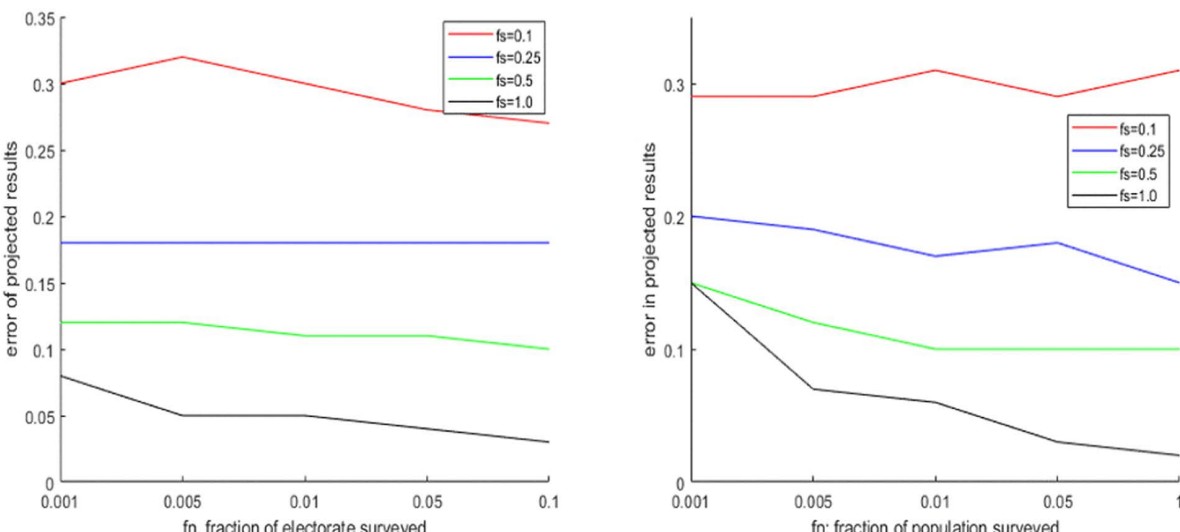

**Fig 7. Change in seat projection errors (Manhattan Distance) due to variation of $f_s$, $f_n$ on elections simulated by DPM (upper part: DPM-6, lower part: DPM-7).**

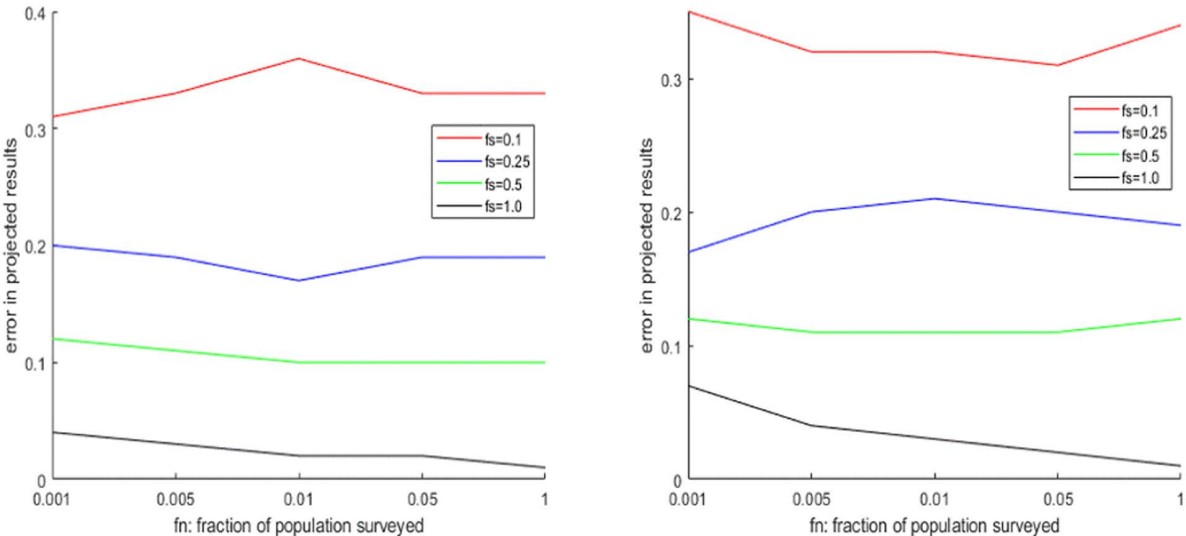

**Fig 8. Change in seat projection errors (Manhattan Distance) due to variation of $f_s$, $f_n$ on elections simulated by PCM (upper part: PCM-6, lower part: PCM-9).**

## 8 Conclusion

Election analysis and result prediction through surveys is a problem that is not only of practical interest to journalists and policymakers, but also of academic interest to political scientists, theoretical computer scientists and statisticians. However, simulations of elections based on detailed voter-centric models are not common. This paper provides an approach that can not only be used to estimate the election outcomes (seat shares of parties) based on their vote shares, but also provides hypothetical results, that could have been realized if the voters were spatially distributed in a different way, or if

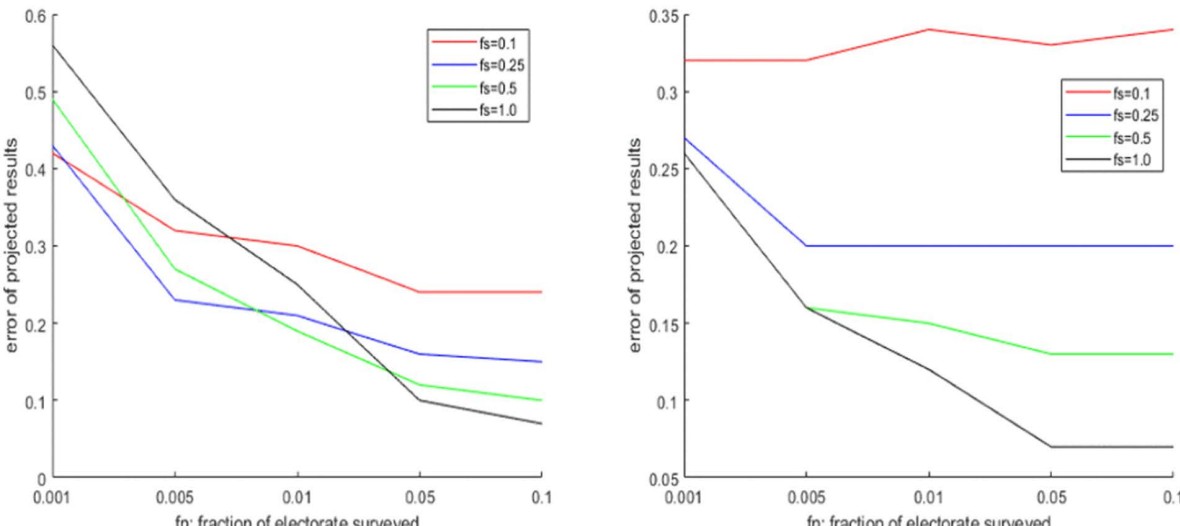

**Fig 9. Change in seat projection errors (Manhattan Distance) due to variation of $f_s$, $f_n$ on elections simulated by SIM (upper part: SIM-1, lower part: SIM-2).**

**Table 7. Accuracy of seat projections due to variation of $f_s$, $f_n$ on some Indian elections.** We report the mean Manhattan Distance in each case, while the probability of accurate projection is shown in brackets.

| $f_s$ | $f_n$ | GJ-17 | GJ-22 | WB-19 | WB-21 |
|---|---|---|---|---|---|
| 0.001 | 0.1 | 0.21 (0.12) | 0.15 (0.19) | 0.17 (0.1) | 0.12 (0.19) |
| 0.001 | 0.25 | 0.12 (0.18) | 0.08 (0.26) | 0.12 (0.21) | 0.07 (0.49) |
| 0.001 | 0.5 | 0.09 (0.33) | 0.05 (0.53) | 0.08 (0.32) | 0.06 (0.43) |
| 0.01 | 0.1 | 0.2 (0.11) | 0.13 (0.22) | 0.19 (0.08) | 0.1 (0.26) |
| 0.01 | 0.25 | 0.12 (0.12) | 0.07 (0.31) | 0.1 (0.28) | 0.07 (0.41) |
| 0.01 | 0.5 | 0.08 (0.31) | 0.05 (0.54) | 0.07 (0.38) | 0.04 (0.68) |

**Table 8. Change in seat projections due to variation of $f_s$, $f_n$ on Indian elections for direct and swing-based projections.**

| Election | $f_n \downarrow$ | Man. Dis. | | | Acc. | | |
|---|---|---|---|---|---|---|---|
| $f_s \rightarrow$ | | 0.1 | 0.25 | 0.5 | 0.1 | 0.25 | 0.5 |
| WB21 (Direct) | 0.001 | 0.13 | 0.07 | 0.05 | 0.15 | 0.44 | **0.6** |
| WB19-WB21 (Swing) | 0.001 | **0.1** | **0.06** | 0.05 | **0.25** | **0.47** | 0.56 |
| WB21 (Direct) | 0.01 | 0.12 | 0.07 | 0.05 | 0.20 | 0.43 | 0.59 |
| WB19-WB21 (Swing) | 0.01 | **0.09** | **0.05** | 0.05 | **0.37** | **0.59** | **0.61** |
| GJ22 (Direct) | 0.001 | 0.14 | 0.08 | 0.06 | 0.14 | 0.20 | **0.50** |
| GJ17-GJ22 (Swing) | 0.001 | **0.10** | **0.07** | 0.06 | **0.28** | **0.35** | 0.45 |
| GJ22 (Direct) | 0.01 | 0.14 | 0.08 | 0.05 | 0.16 | 0.27 | **0.61** |
| GJ17-GJ22 (Swing) | 0.01 | **0.09** | **0.07** | 0.05 | **0.33** | **0.43** | 0.52 |

the community-party relationships were different. Such analysis can lead to understanding the fairness and robustness of a given districting system or voting policy, such as plurality or first-past-the-post. Additionally, we also look into swing of votes across successive elections, and propose two new models that are far more realistic than the uniform or proportional swing models, as it allows for the possibility that even a party that loses votes overall can gain new seats. We also simulate election surveys, and show the complex relationships between accuracy of projections and district coverage or sample size of respondents. We also find that if the district coverage or sample size is small, then better projection results can be obtained by estimating the swing matrix with respect to the previous election, than directly trying to estimate the seat share of the current election. These results can provide directions to polling agencies regarding their sampling and querying approach. Our results are validated on actual elections held in different states of India.

## Supporting information

**S1 File. Additional analyses and results.**
(PDF)

## Acknowledgments

Adway Mitra thanks Indian Institute of Technology Kharagpur for partial support for this research.

## Author contributions

**Conceptualization:** Adway Mitra.

**Data curation:** Adway Mitra.

**Formal analysis:** Adway Mitra.

**Investigation:** Adway Mitra.

**Methodology:** Adway Mitra.

**Resources:** Adway Mitra.

**Software:** Adway Mitra.

**Validation:** Adway Mitra.

**Visualization:** Adway Mitra.

**Writing – original draft:** Adway Mitra.

**Writing – review & editing:** Adway Mitra.

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
