## [Decision Letter · Decision Letter 0]

5 Nov 2025

Dear Dr. Mitra,

Thank you for submitting your manuscript to PLOS ONE. After careful consideration, we feel that it has merit but does not fully meet PLOS ONE’s publication criteria as it currently stands. Therefore, we invite you to submit a revised version of the manuscript that addresses the points raised during the review process.

**ACADEMIC EDITOR:**plosone@plos.org . A rebuttal letter that responds to each point raised by the academic editor and reviewer(s). You should upload this letter as a separate file labeled 'Response to Reviewers'.A marked-up copy of your manuscript that highlights changes made to the original version. You should upload this as a separate file labeled 'Revised Manuscript with Track Changes'.An unmarked version of your revised paper without tracked changes. You should upload this as a separate file labeled 'Manuscript'.

We look forward to receiving your revised manuscript.

Kind regards,

Omar El Deeb

Academic Editor

PLOS ONE

Journal Requirements:

3. We note you have included a table to which you do not refer in the text of your manuscript. Please ensure that you refer to Table 13 in your text; if accepted, production will need this reference to link the reader to the Table.

Reviewers' comments:

Reviewer's Responses to Questions

**Comments to the Author**

1. Is the manuscript technically sound, and do the data support the conclusions?

Reviewer #1: Yes

Reviewer #2: Partly

2. Has the statistical analysis been performed appropriately and rigorously?

Reviewer #1: Yes

Reviewer #2: Yes

3. Have the authors made all data underlying the findings in their manuscript fully available?

Reviewer #1: Yes

Reviewer #2: Yes

4. Is the manuscript presented in an intelligible fashion and written in standard English?

Reviewer #1: Yes

Reviewer #2: No

Reviewer #1: This manuscript presents a technically ambitious and original contribution to the study of district-based elections. It integrates agent-based modeling, Monte Carlo simulation, and Dirichlet Process–based statistical inference to examine how spatial heterogeneity, social identity, and temporal vote swings shape electoral outcomes. The study’s main contribution lies in introducing Dirichlet-based swing models for electoral dynamics, which extend existing statistical approaches to vote transitions by offering a more nuanced treatment of spatial and temporal heterogeneity. This interdisciplinary framework bridges political science, statistics, and computational modeling, providing a scalable method capable of simulating real-world multiparty elections with greater realism than conventional swing models. Moreover, it opens promising avenues for integrating survey simulations and predictive analytics in heterogeneous societies, marking an innovative direction with strong cross-disciplinary potential.

Reviewer #2: In this work, the authors compare multiple election models. Finally, they use the models to reconstruct the election results in India. The authors propose two new models. The work is very extensive and references numerous works of literature. It would greatly benefit from the addition of graphics comparing models, model features, etc., as well as from greater emphasis on where the author's model is discussed. Other methods that would improve the readability of the work were also highly recommended.Works that compare methods/models are very necessary, but in such works, their usefulness to the reader is even more important. The authors should overcomplicate the work, perhaps by shortening it and making it more attractive with graphics, but also by making it more user-friendly.

**Do you want your identity to be public for this peer review?** For information about this choice, including consent withdrawal, please see our Privacy Policy

Reviewer #1: No

Reviewer #2: No

---

## [Author Response · Author response to Decision Letter 1]

11 Jan 2026

Comments by Academic Editor:

Comment: I recommend that the authors take into account the remarks of reviewer 2, especially remarks related to introducing graphics that may simplify cross-comparisons between models.

Response: I already had 3 figures that demonstrate the spatial maps of district-based elections simulated by the models discussed. In addition, I have added plots to compare vote share vs seat share (Figures 4 and 5) in the proposed swing models. I have also replaced the tabular presentation of the results related to the impact of ‘district coverage’ and ‘person coverage’ parameters on the accuracy of election surveys, using plots for the same (Figure 6-8). I hope these graphics will simplify the visualization of the results.

Comment: Regarding the issue of social identity and voting behavior, there are few studies that the authors may find useful - I send them few suggestions, in case they think they have relevance in this context of the literature review of this study (doi.org/10.1371/journal.pone.0331959 & doi.org/10.1016/j.physa.2023.128675).

Response: I thank the editor for these suggestions; they are indeed relevant to this study. I have cited both of these papers, and also discussed the position of this paper with respect to them lines (215-218) and (316-319).

Comment: Another issue raised by reviewers is the length of the manuscript. I suggest a concise writing and that redundancies be summarized to make the paper more readable and accessible for a broad audience.

Response: I have removed some experimental results which I feel are less important for the main premise of the paper to an “Appendix” section. These include the complete tabular data related to simulation of swings by the proposed DSM and DSMM models (graphical description of the results has already been provided in Figures 4 and 5 in the main text). The results about spatial variation of the vote shares, and the full tabular data related to the variation of survey results based on survey parameters have also been moved to the “Appendix”. I am not sure if PLoS One allows “Appendix” sections – if not, this section can be dropped altogether. Additionally, I have modified the text to make it more concise wherever possible.

Comments by Reviewer 1: This manuscript presents a technically ambitious and original contribution to the study of district-based elections. It integrates agent-based modeling, Monte Carlo simulation, and Dirichlet Process–based statistical inference to examine how spatial heterogeneity, social identity, and temporal vote swings shape electoral outcomes. The study’s main contribution lies in introducing Dirichlet-based swing models for electoral dynamics, which extend existing statistical approaches to vote transitions by offering a more nuanced treatment of spatial and temporal heterogeneity. This interdisciplinary framework bridges political science, statistics, and computational modeling, providing a scalable method capable of simulating real-world multiparty elections with greater realism than conventional swing models. Moreover, it opens promising avenues for integrating survey simulations and predictive analytics in heterogeneous societies, marking an innovative direction with strong cross-disciplinary potential.

Response: I thank Reviewer 1 for their positive assessment of the work.

Comments by Reviewer 2: In this work, the authors compare multiple election models. Finally, they use the models to reconstruct the election results in India. The authors propose two new models. The work is very extensive and references numerous works of literature. It would greatly benefit from the addition of graphics comparing models, model features, etc., as well as from greater emphasis on where the author's model is discussed. Other methods that would improve the readability of the work were also highly recommended. Works that compare methods/models are very necessary, but in such works, their usefulness to the reader is even more important. The authors should overcomplicate the work, perhaps by shortening it and making it more attractive with graphics, but also by making it more user-friendly.

Response: I thank Reviewer 2 for their assessment of the work and the suggestions. I have addressed the suggestions in my response to the Academic Editor.

Other issues: I have made sure that all tables and figures provided have been referenced in the text. I have used the PLoS One Latex Template, and followed the formatting instructions as far as possible.

---

## [Decision Letter · Decision Letter 1]

3 Feb 2026

Dear Dr. Mitra,

Thank you for submitting your manuscript to PLOS ONE. After careful consideration, we feel that it has merit but does not fully meet PLOS ONE’s publication criteria as it currently stands. Therefore, we invite you to submit a revised version of the manuscript that addresses the points raised during the review process.

We look forward to receiving your revised manuscript.

Kind regards,

Omar El Deeb

Academic Editor

PLOS One

Journal Requirements:

Reviewers' comments:

Reviewer's Responses to Questions

**Comments to the Author**

Reviewer #1: All comments have been addressed

Reviewer #2: (No Response)

2. Is the manuscript technically sound, and do the data support the conclusions?

Reviewer #1: Yes

Reviewer #2: Yes

3. Has the statistical analysis been performed appropriately and rigorously?

Reviewer #1: Yes

Reviewer #2: Yes

4. Have the authors made all data underlying the findings in their manuscript fully available?

Reviewer #1: Yes

Reviewer #2: Yes

5. Is the manuscript presented in an intelligible fashion and written in standard English?

Reviewer #1: Yes

Reviewer #2: Yes

Reviewer #1: Overall, the author has adequately and constructively addressed the main points raised in the previous round of review, particularly those concerning clarity, presentation, and comparative readability of the models.

First, in response to the request for improved graphical presentation and clearer cross-model comparisons, the revised manuscript introduces several new figures and replaces parts of the earlier tabular presentation with visualizations. In particular, the addition of plots comparing vote share and seat share under the proposed swing models (Figures 4 and 5), as well as graphical illustrations of survey parameter effects (Figures 6–8), directly responds to the editor’s and Reviewer 2’s concerns. These additions substantially improve accessibility and make it easier for readers to compare model behavior across parameter settings

Second, regarding engagement with the literature on social identity and voting behavior, the author has explicitly incorporated the suggested references and clarified how the present work relates to them. The revisions include additional citations and brief positioning statements in the literature review and discussion sections, which sufficiently acknowledge and contextualize prior work without overextending the scope of the paper.

Third, the concern about manuscript length and redundancy has been addressed in a reasonable manner. The author has shortened the main text by moving less central numerical results and extensive tables to an Appendix, while retaining the core analytical narrative and visual summaries in the main manuscript. This improves readability and aligns the paper more closely with PLOS ONE’s emphasis on clarity and accessibility. Although the author notes some uncertainty about the journal’s handling of appendices, the effort to streamline the manuscript is evident and effective.

Finally, no new substantive issues appear to have been introduced in the revision. Formatting, figure referencing, data availability statements, and ethical declarations are now complete and consistent with journal requirements.

Reviewer #2: The authors created graphs comparing the described and simulated models, but the graphs are of very poor quality. They are very difficult to read; they seem blurry. In order to make things easier for the reader, in the previous comments I suggested a flowchart for the models. It may not have been described in great detail, but in my opinion, a block diagram was best, showing which model contains which assumptions, so that it would be easy to understand the differences between the models.

**Do you want your identity to be public for this peer review?** For information about this choice, including consent withdrawal, please see our Privacy Policy

Reviewer #1: No

Reviewer #2: No

---

## [Author Response · Author response to Decision Letter 2]

11 Feb 2026

I thank the Reviewers and the Academic Editor for their time and effort in reviewing my paper. I have highlighted the newly added text in green colour, and indicated removed text by striking through.

Response to Reviewer 1: I thank Reviewer 1 for their favorable assessments.

Comments by Reviewer 2: The authors created graphs comparing the described and simulated models, but the graphs are of very poor quality. They are very difficult to read; they seem blurry. In order to make things easier for the reader, in the previous comments I suggested a flowchart for the models. It may not have been described in great detail, but in my opinion, a block diagram was best, showing which model contains which assumptions, so that it would be easy to understand the differences between the models.

Response to Reviewer 2: I have replaced some of the previous images to improve their sharpness and quality. I have also added two new figures – Fig 1: a block diagram showing the workflow of the paper, indicating the roles of the different models including their inputs and outputs, and Fig 2: a table listing the different models along with their inputs, outputs, parameters and assumptions.

---

## [Editor Report · Decision Letter 2]

15 Feb 2026

Dirichlet-Swing: understanding spatio-temporal aspects of political elections in heterogeneous societies through agent-based simulation

PONE-D-25-46601R2

Dear Dr. Mitra,

We’re pleased to inform you that your manuscript has been judged scientifically suitable for publication and will be formally accepted for publication once it meets all outstanding technical requirements.

Kind regards,

Omar El Deeb

Academic Editor

PLOS One
---

## [Editor Report · Acceptance letter]

PONE-D-25-46601R2

PLOS One

Dear Dr. Mitra,

I'm pleased to inform you that your manuscript has been deemed suitable for publication in PLOS One. Congratulations! Your manuscript is now being handed over to our production team.

Kind regards,

on behalf of

Dr. Omar El Deeb

Academic Editor

PLOS One